

# From eggs to bites: do ovitrap data provide reliable estimates of *Aedes albopictus* biting females?

Mattia Manica[1,2], Roberto Rosà[2], Alessandra della Torre[1] and Beniamino Caputo[1]

[1] Dipartimento di Sanitá Pubblica e Malattie Infettive, Laboratory affiliated to Istituto Pasteur Italia – Fondazione Cenci Bolognetti, Sapienza University of Rome, Rome, Italy

[2] Dipartimento di Biodiversità ed Ecologia Molecolare/Centro Ricerca e Innovazione, Fondazione Edmund Mach, San Michele all'Adige, Italy

Corresponding author
Beniamino Caputo,
beniamino.caputo@uniroma1.it

## ABSTRACT

**Background.** *Aedes albopictus* is an aggressive invasive mosquito species that represents a serious health concern not only in tropical areas, but also in temperate regions due to its role as vector of arboviruses. Estimates of mosquito biting rates are essential to account for vector-human contact in models aimed to predict the risk of arbovirus autochthonous transmission and outbreaks, as well as nuisance thresholds useful for correct planning of mosquito control interventions. Methods targeting daytime and outdoor biting *Ae. albopictus* females (e.g., Human Landing Collection, HLC) are expensive and difficult to implement in large scale schemes. Instead, egg-collections by ovitraps are the most widely used routine approach for large-scale monitoring of the species. The aim of this work was to assess whether ovitrap data can be exploited to estimate numbers of adult biting *Ae. albopictus* females and whether the resulting relationship could be used to build risk models helpful for decision-makers in charge of planning of mosquito-control activities in infested areas.

**Method.** Ovitrap collections and HLCs were carried out in hot-spots of *Ae. albopictus* abundance in Rome (Italy) along a whole reproductive season. The relationship between the two sets of data was assessed by generalized least square analysis, taking into account meteorological parameters.

**Result.** The mean number of mosquito females/person collected by HLC in 15′ (i.e., females/HLC) and the mean number of eggs/day were $18.9 \pm 0.7$ and $39.0 \pm 2.0$, respectively. The regression models found a significant positive relationship between the two sets of data and estimated an increase of one biting female/person every five additional eggs found in ovitraps. Both observed and fitted values indicated presence of adults in the absence of eggs in ovitraps. Notably, wide confidence intervals of estimates of biting females based on eggs were observed. The patterns of exotic arbovirus outbreak probability obtained by introducing these estimates in risk models were similar to those based on females/HLC ($R0 > 1$ in 86% and 40% of sampling dates for Chikungunya and Zika, respectively; $R0 < 1$ along the entire season for Dengue). Moreover, the model predicted that in this case-study scenario an $R0 > 1$ for Chikungunya is also to be expected when few/no eggs/day are collected by ovitraps.

**Discussion.** This work provides the first evidence of the possibility to predict mean number of adult biting *Ae. albopictus* females based on mean number of eggs and to compute the threshold of eggs/ovitrap associated to epidemiological risk of arbovirus

transmission in the study area. Overall, however, the large confidence intervals in the model predictions represent a caveat regarding the reliability of monitoring schemes based exclusively on ovitrap collections to estimate numbers of biting females and plan control interventions.

## INTRODUCTION

*Aedes albopictus* (Skuse) (Diptera: Culicidae) is an aggressive daytime biting invasive mosquito species (*Hawley, 1988*) which represents a serious health concern not only in tropical areas, but also in temperate regions of Europe, US and China where it is now well established (*Medlock et al., 2015*). In fact, the species is a competent vector for many arboviruses (*Gratz, 2004*), such as the most recent pandemic Zika virus (*Di Luca et al., 2016*), and has been responsible for large Chikungunya virus epidemics in Indian Ocean islands and in India (*Higgs, 2006*; *Enserink, 2006*; *Roth et al., 2014*). In Europe, it was responsible for the first outbreak of an exotic arbovirus (i.e., >200 confirmed Chikungunya cases in Ravenna Province, north-east Italy in 2007) and of the transmission of autochthonous cases of Dengue and Chikungunya in France and Croatia in more recent years (*Rezza et al., 2007*; *Angelini et al., 2007*; *Gjenero-Margan et al., 2011*; *Grandadam et al., 2011*; *Delisle et al., 2015*; *Succo et al., 2016*).

Estimates of mosquito biting rates are essential to account for vector-human contact in models aiming at predicting the risk of autochthonous transmission and outbreaks of mosquito-borne diseases, as well as mosquito nuisance. These estimates can be obtained by collecting mosquitoes on human volunteers (i.e., human landing collection, HLC), a very labour-intensive process, unethical in areas of proven disease transmission (*Silver, 2008*). Other methods targeting biting females of daytime outdoor biting species (e.g., BG-sentinel traps for *Ae. albopictus*) are expensive and difficult to implement in large scale schemes. Thus, models aimed to predict the risk of autochthonous transmission and outbreaks of arbovirus by *Ae. albopictus* are constrained by the difficulty to obtain fine-scale entomological data.

On the other hand, the most widely available entomological data for *Ae. albopictus* come from egg-collection by ovitraps, a routine large-scale monitoring approach. This has been largely exploited by public administrations to survey the species abundance, due to its limited implementing costs (*ECDC, 2012*). The use of egg abundance in risk models can be convenient, provided this can be proved to be a good predictor of biting adults. However, the relationship between mosquito eggs and biting females is not straightforward (*Qiu et al., 2007*; *ECDC, 2012*) and may be differently affected by climatic (e.g., temperature, rainfall, wind; *Hawley, 1988*; *Waldock et al., 2013*; *Vallorani et al., 2015*), ecological (e.g., number of alternative oviposition sites *Davis et al., 2015*) and demographic (e.g., human and alternative hosts densities) factors.

As of today, no studies have attempted to quantitatively predict numbers of adult biting *Aedes* females from ovitrap data, although a study from Indonesia showed a positive correlation between eggs in ovitrap and number of host-seeking *Aedes aegypti* females in BG-sentinel traps (*Tantowijoyo et al., 2016*). The aims of the present study were to (i) investigate the relationship between the mean number of human-biting *Ae. albopictus* females and number of eggs in ovitraps along the mosquito reproductive season and (ii) assess the accuracy of this relationship. An accurate prediction of numbers of adult biting females from ovitrap data would in fact provide decision-makers in charge of planning of mosquito-control activities with a straightforward measure of high mosquito densities, associated to higher nuisance, as well as higher risk of arbovirus outbreaks. In order to achieve these goals, we carried out parallel ovitrap and human landing collections in two hot-spots of high *Ae. albopictus* abundance in Rome (Italy) and assessed the relationship between the two sets of data by regression analysis.

## MATERIALS & METHODS

### Study sites

Human Landing Collections (HLC) and ovitrap collections were carried out from July 21th to October 31th 2014 in two *Ae. albopictus* heavily infested study sites (∼1-hectar each) inside the metropolitan area of Rome (Italy), at about 400 m distance from each other: the botanical garden inside the campus of La Sapienza University of Rome (Site A, 41°54′12.6″N and 12°30′59.7″E; see *Cianci et al., 2015*) and the enclosed garden of the Institute of Anatomy (Site B, 41°54′23.32″N and 12°30′57.35″E; see *Caputo et al., 2012*).

### Mosquito collections

Human Landing Collections were performed three days per week (i.e., on Monday, Wednesday and Friday) by two qualified operators in two outdoor spots located at a distance of approximately 100 m within each study site. The operators gave their consent to carry out HLC after being informed of potential risks. At planned day, collections started 1 h before sunset and finished within 30 min. Each HLC (i.e., a single collection made by a single operator in one spot) lasted for 15 min; after rotating between spots within the site, operators moved to the second site. In the following day of collection, the first site sampled was the second one sampled in previous collection day. In case of rain immediately before or during HLC time, collections were postponed to the next scheduled day. During each HLC, the operator seated exposing a ∼4,200 cm$^2$ naked area in one foreleg. Biting female mosquitoes were killed with a racket zapper as soon as they landed on the skin. Killed mosquitoes were identified and counted directly in the field.

Egg collections were carried out by ovitraps filled with 300 ml water and internally lined with a germination paper on which mosquito females lay their eggs (*Velo et al., 2016*). Ten ovitraps were positioned in site A and five in site B (this difference in number of ovitraps is due to lack of open space derived by the presence of a large building in site B). In the same day of HLC, operators collected germination papers in sealed plastic bags, emptied ovitraps, and replenished them with tap water. Egg counting was carried out under a stereomicroscope in the laboratory. Each month, approximately 1/10 of collected eggs
were hatched and reared to the adult stage in order to confirm exclusive presence of *Ae. albopictus*.

In view of the following considerations we assume that removing *Ae. albopictus* adult females and their eggs from the field doesn't significantly affect the mosquito population size and temporal dynamics: (i) collections were carried out in typical hot-spots of high *Ae. albopictus* density (*Manica et al., 2016*) in heavily infested areas (*Marini et al., 2010*; *Cianci et al., 2015*; *Caputo et al., 2015*); (ii) after the arrival in an infested area a human host can attract all the females present within a radius of only 4–7 m in 15′ HLC (*Mogi & Yamamura, 1981*); (iii) the time required by HLC represents only a small fraction of the overall female daily biting activity (*Hawley, 1988*); (iv) the number of ovitraps employed is to be considered negligible compared to number of potential natural breeding sites in the study sites (e.g., catch basins, vases, pots, flowerpot saucers).

## Meteorological data

Meteorological data (i.e., hourly records of temperature at 2 m from ground, wind speed and precipitation) were obtained by the opendata archive of the ''Ministero delle Politiche Agricole, Alimentari e Forestali'' (weather station Roma Collegio Romano https://www.politicheagricole.it/flex/cm/pages/ServeBLOB.php/L/IT/IDPagina/7012, accessed 2 June 2015). Meteorological data were aggregated to obtain the following variables of interest:

- daily average wind speed, average temperature and total mm of rainfall;
- a binary rainfall index indicating the occurrence of rainfall during the day;
- average temperature and accumulated mm of rainfall recorded over one, two, three weeks prior to collection day.

## Statistical analysis

All analyses were carried out using the software R version 3.3.1 (*R Core Team, 2016*) and the packages nlme (*Pinheiro et al., 2016*), MuMIn (*Barton, 2016*), AICcmodavg (*Mazerolle, 2016*) and ggplot2 (*Wickham, 2009*).

A Pearson correlation between the mean number of female/site/day (i.e., the mean number of biting *Ae. albopictus* females collected by the two operators in the two spots within a site in a single day) and the mean number of eggs/site/day at lag 0 (i.e., the mean number of eggs from each ovitrap within each site divided by the number of days the ovitrap was active) was computed.

### Basic estimate of biting females based on mean number of egg/day in ovitrap (Model-I)

This relationship was tested by means of regression analysis also accounting for meteorological variables that could affect HLC sampling. Response variable was the mean number of female/site/day (i.e., the mean number of biting *Ae. albopictus* females collected by the two operators in the two spots within a site in a single day). Explanatory variables were site, mean number of eggs/site/day at lag 0 (i.e., the mean number of eggs from each ovitrap within each site divided by the number of days the ovitrap was active), mean number of eggs/site/day at lag 1 (i.e., the mean number of eggs/site/day in the seven days preceding HLC sampling), the mean number of eggs/site/day at lag 2 (i.e., the mean number

of eggs/site/day from 7 to 14 days preceding HLC sampling). The choice of lag 0, 1 and 2 was based on: (i) the mean time from egg oviposition to first blood-meal, which during the summer months in temperate areas is <14 days (B Caputo, pers. obs., 2014), and (ii) the fact that routine ovitrap surveillance in large-scale monitoring schemes is usually carried on a weekly base, at least in Italy (*ISS, 2016*).

In addition, some explanatory variables were included, i.e., meteorological variables recorded on the day of HLC sampling such as the precipitation occurrence (yes or no) and the average daily values for wind speed, temperature and temperature quadratic term. Temperature and wind data were centred (subtracted its mean) to help interpretation of results (*Schielzeth, 2010*). Due to irregularly observed data and the longitudinal structure of the data, a continuous auto-regressive correlation structure of order 1 was considered in the model. The resulting model was fitted using the generalized least squared method by maximizing the restricted log-likelihood (REML). Model assumptions were verified by checking the model normalized residuals for any pattern or dependency. This model, hereafter-defined "full model", including all the ecologically relevant parameters available, was used to generate a set of all plausible sub-models. The model considering the temperature quadratic term included also the linear one. A multi-model selection approach (*Burnham & Anderson, 2002*) was then employed to compare all models in the set. Models were ranked by AICc (*Burnham & Anderson, 2002*) using maximum likelihood estimation (ML) (*Faraway, 2006*). Results of the ranking process were used to calculate weights and the relative importance for each variable by summing the Akaike weights for each model that contains the parameter of interest. The model having the lowest AIC was then selected and refitted using REML Model, performance was assessed using in-sample errors by computing the root mean squared error (RMSE), which represents the sample standard deviation of the differences between predicted values and observed values and could be interpreted as an estimation of the standard deviation of the unexplained variance. Pearson correlation between observed and fitted mean values was also computed.

### Improved estimate of biting females based on mean number of egg/day in ovitrap (Model-II)

Following the same approach, we built a new regression model aiming at improving the basic prediction of biting females obtained from Model I where only egg counts and short-term meteorological variables were considered. Specifically, we added average values for meteorological variables (temperature and precipitation) computed for a longer period preceding HLC sampling (till three weeks before) in order to take into account the effect of climatic variables not only on HLC sampling, due to mosquito activity, but also on mosquito population dynamics. Explanatory variables were the same used in Model-I: site, mean number of eggs/site/day (only at lag 0), the precipitation occurrence (yes or no) and the average values for wind speed and temperature quadratic term recorded on the day of collection. In addition, in this case, the average daily temperature and accumulated precipitation, with their quadratic terms, recorded over the previous one, two, three weeks were also included as explanatory variables. Again, temperature, wind and rainfall variables were centred and a continuous auto-regressive correlation structure of order 1 was considered. A

set of plausible sub-models was then generated. The model set was tailored in order to retain models considering at most three meteorological variables (one for temperature, one for rainfall and one for wind) in order to avoid collinearity among meteorological explanatory variables. Models considering the quadratic terms included also the corresponding linear one. All models in the set were then compared and ranked by AICc (*Burnham & Anderson, 2002*) using ML estimation (*Faraway, 2006*). The model having the lowest AIC was then selected and refitted using REML. RMSE and Pearson correlation between observed and fitted mean values were computed. Collinearity was investigated using the function corvif (*Zuur et al., 2009*). During the model validation process, a simulation study was carried out to assess how the relationship between the mean number of egg/day in ovitraps and biting females from HLC, obtained from the best Model II, is influenced by the number of ovitraps considered. To test this, Model-II was re-fitted on simulated subsets of the original dataset; precisely, subsets were simulated by fixing at each step the number of ovitraps included in the analysis (from one to 15 traps, that is the actual number used in the best Model II) and then resampling with replacement (1,000 times each step) the number of ovitraps to be considered. Model-II was re-fitted on every subset in order to obtain mean values and 95% confidence intervals for the parameters of interest (i.e., the estimated value of the mean number/eggs/day parameter, its significance, the RMSE and the Pearson correlation) for each fixed number of ovitraps.

### Basic reproduction number and outbreak probability of exotic arbovirus

The basic reproduction number ($R_0$) for mosquito-borne arboviruses such as Chikungunya, Dengue and Zika virus can be calculated from densities of human and mosquito populations and several epidemiological parameters according to the following formula $R_0 = R_0^{HV} R_0^{VH}$ (*Smith et al., 2012*). Symbols, interpretations, values and literature references for each parameter are reported in the Table 1. Specifically, $R_0^{HV} = \frac{k \chi_V}{\gamma} \frac{V}{H} \frac{\omega_V}{\omega_V + m}$ could be interpreted as the product of the number of infectious mosquitoes generated from an infectious human while $R_0^{VH} = \frac{k \chi_H}{m}$ as the number of infectious humans generated by the infectious mosquitoes surviving the extrinsic incubation period. When $R_0 < 1$ (epidemic threshold), the probability of observing sustained arbovirus transmission after importation of a case is negligible. When $R_0 > 1$, the outbreak probability is given by the following formula: $p = 1 - \frac{R_0^{VH} + 1}{R_0^{VH}(R_0^{HV} + 1)}$.

HLC-observed data and HLC-predicted values obtained from Model-2, multiplied by a correction factor $x$ as in *Carrieri et al. (2012b)*, were used to estimate the number of bites on human per mosquito ($kV/H$).

## RESULTS

### Ovitrap and HLC collections

A total of 5,678 biting *Ae. albopictus* adult females and 25,120 *Ae. albopictus* eggs were collected. The mean number of females/person collected by HLC in 15′ (hereafter females/HLC) was 20.8 ($\pm$0.9 SE) and 17.1 ($\pm$0.9 SE) in Site-A and in Site-B, respectively. The maximum number of females/HLC was 47 in Site-A and 45 in site-B. The mean number of eggs/day was 35.6 ($\pm$3.4 SE) and 40.7 ($\pm$2.4 SE) in Site-A and Site-B, respectively. The

Manica et al. (2017), PeerJ, DOI 10.7717/peerj.2998

**Table 1  Epidemiological parameters.** Symbols, values and references for the parameters used.

| Parameter | Description | CHIKV | | DENV | | ZIKAV | |
|---|---|---|---|---|---|---|---|
| | | Value (range) | Reference | Value (range) | Reference | Value | Reference |
| $k$ | Human biting rate (the number of bites to humans per mosquito per day) | 0.09 (0.05–0.16) | *Poletti et al. (2011)* | 0.09 (0.05–0.16) | *Poletti et al. (2011)* | 0.09 (0.05–0.16) | *Poletti et al. (2011)* |
| $m$ | Mortality rate ($1/g$ = average mosquito life-span in days) | Function (Temperature) | *Poletti et al. (2011)* | Function (Temperature) | *Poletti et al. (2011)* | Function (Temperature) | *Poletti et al. (2011)* |
| $\chi_H$ | Susceptibility to infection of humans, transmission efficiency from an infected mosquito to human | 65% (50%–80%) | *Dumont, Chiroleu & Domerg (2008)* | 31% (10%–50%) | *Manore et al. (2014)* | 50% (1%–100%) | *Wong et al. (2013)* and *Chouin-Carneiro et al. (2016)* |
| $\chi_V$ | Susceptibility to infection of mosquito, transmission efficiency from an infected human to mosquito | 85% (70%–100%) | *Talbalaghi et al. (2010)* and *Vega-Rua et al. (2013)* | 31% (10%–50%) | *Manore et al. (2014)* | 50% (0.8%–100%) | *Wong et al. (2013)* and *Chouin-Carneiro et al. (2016)* |
| $1/\omega_V$ | Length of extrinsic incubation period | 2.5 (2–3) days | *Dumont, Chiroleu & Domerg (2008)* and *Dubrulle et al. (2009)* | 10 (7–14) days | *Manore et al. (2014)* | 10.5 (7–14) days | *Guzzetta et al. (2016a)* |
| $1/\gamma$ | Infectious period in human hosts | 4.5 (2–7) days | *Parola et al. (2006)* and *Dumont, Chiroleu & Domerg (2008)* | 6 (3–7) days | *Manore et al. (2014)* | 5.8 (4–7) days | *Guzzetta et al. (2016a)* |
| $X$ | Correction factor | 0.101 | *Carrieri et al. (2012b)* | 0.101 | *Carrieri et al. (2012b)* | 0.101 | *Carrieri et al. (2012b)* |
| $kV/H$ | Ratio of mosquito per human | Time dependent | Observed by human landing collection | Time dependent | Observed by human landing collection | Time dependent | Observed by human landing collection |

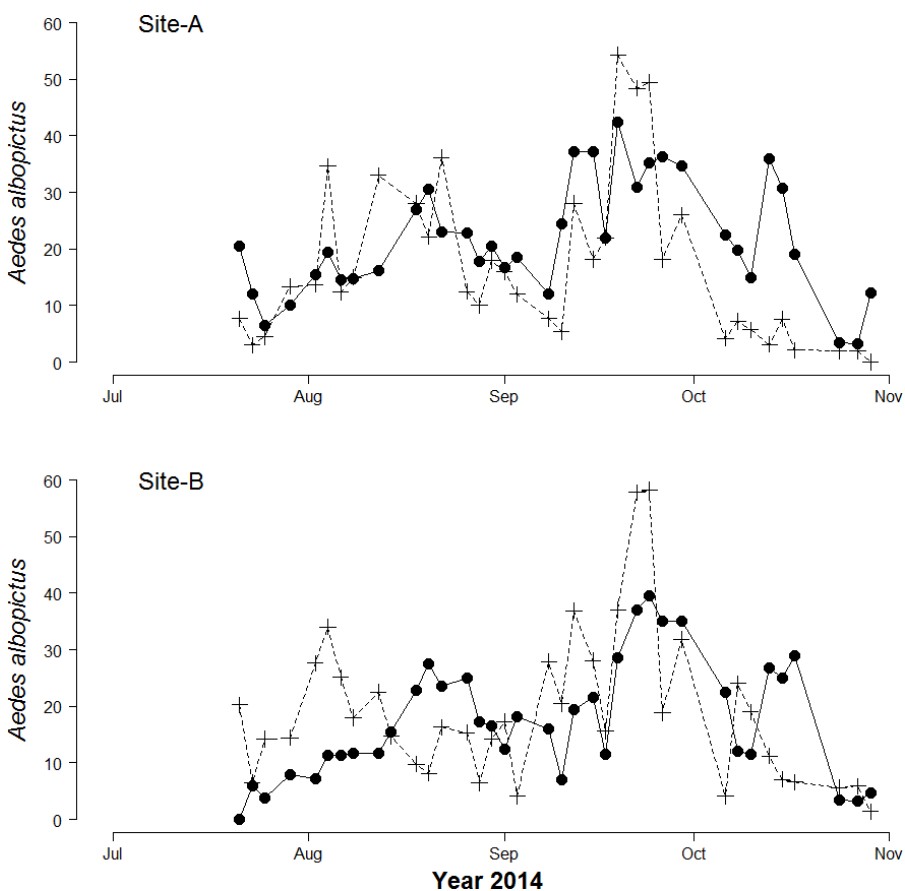

**Figure 1** **Seasonal patterns of *Aedes Albopictus* eggs and adults.** Seasonal patterns of eggs and adults per site per day in botanical garden (Site-A) and the enclosed garden of the Institute of Anatomy (Site-B) in Sapienza University, Rome, Italy. Dots represent mean biting adults. Crosses represent mean eggs/day.

maximum number of eggs collected in one ovitrap in a single sampling was 288 in Site-A and 300 in Site-B. No eggs were found in 109 out of 644 ovitrap collections (16.9%). A bimodal temporal pattern of egg and adult abundance, consistent with the pattern observed in previous years (*Manica et al., 2016*), was observed in both study sites (Fig. 1). A significant Pearson correlation was found between the mean number of female/site/day and the mean number of eggs/site/day at lag 0 ($r = 0.47$, df $= 71$, *p*-values $= <0.0001$).

## Basic estimate of biting females based on mean number of egg/day in ovitrap

Results of regression analysis carried out to estimate biting females based on mean number of egg/day accounting for meteorological variables that could affect HLC sampling—show that the model with lowest AIC had as explanatory variables the mean number of eggs/site/-day at lag 0 and average daily wind measured at day of sampling (Model-I; Table 2; Table S1). The estimated parameter for the continuous AR1 correlation is 0.85. The model-averaged importance of terms computed after the multi-model selection process (192 models) are mean number/eggs/day lag 0 (0.81) and temperature (0.52). Other explanatory variables

**Table 2 Coefficient and statistics of the parameters for Model-I.** Coefficient and statistics of the parameters for the best (lowest AIC) generalized least square model with continuous AR1 correlation structure analysing the relationship between the mean numbers of biting *Ae. albopictus* females/site/day and the mean number eggs/site/day.

| Coeff. | Value | SE | *T*-value | *p*-value |
|---|---|---|---|---|
| Intercept | 14.719 | 2.493 | 5.904 | <0.0001 |
| Mean number of eggs/site/day | 0.233 | 0.071 | 3.280 | 0.0016 |
| Wind | −1.855 | 1.221 | −1.519 | 0.1334 |

**Table 3 Coefficient and statistics of the parameters for Model-II.** Coefficient and statistics of the parameters for the best (lowest AIC) generalized least square model with continuous AR1 correlation structure analysing the relationship between the mean numbers of biting *Ae. albopictus* females/site/day and the mean number eggs/site/day accounting for the lagged effects of meteorological variables.

| Coeff. | Value | SE | *T*-value | *p*-value |
|---|---|---|---|---|
| Intercept | 20.109 | 2.438 | 8.247 | <0.0001 |
| Mean number of eggs/site/day | 0.245 | 0.073 | 3.337 | 0.0014 |
| Temp | −0.891 | 0.471 | −1.891 | 0.0630 |
| Temp$^2$ | −0.289 | 0.086 | −3.348 | 0.0013 |
| Rain 2 week lag | −0.141 | 0.081 | −1.739 | 0.0867 |
| Wind | −2.943 | 1.321 | −2.228 | 0.0293 |
| Site-B | −4.648 | 2.620 | −1.774 | 0.0807 |

with values <0.50 are mean number/eggs/day lag 1 (0.50), wind (0.44), rain occurrence (0.37), site (0.37), mean number/eggs/day lag 2 (0.35), and temperature$^2$ (0.28). A positive relationship between the mean numbers of females/HLC and the mean numbers eggs/site/day is observed (Fig. 2A). The estimated coefficient for the mean number of eggs/site/day is 0.233 However, Model-I does not satisfactory explain the variability of the collected number of adult females (Pearson correlation = 0.53; RSME = 8.9; Fig. 2B) and only partially describes the observed temporal pattern of biting females (Figs. 2C and 2D).

## Improved estimate of biting females based on mean number of egg/day in ovitrap

In order to improve the accuracy of estimates, meteorological variables that may affect the mosquito population dynamics were added to Model-I. After model ranking (Table S2), the explanatory variables of the model with lowest AIC (Model-II) are the mean number of eggs/site/day, the wind, the mean temperature in the day when HLCs were carried out and its quadratic term, the mean rainfall during two weeks before HLC and the two Sites (Table 3). The parameter estimate for the continuous AR1 correlation is 0.70. As for Model-I, a positive relationship between the mean numbers of females/HLC and the mean numbers eggs/site/day is observed; the estimated coefficient for the mean number of eggs/site/day is 0.245 (Fig. 3A). Compared to Model-I, Model-II better explains the variability of the collected number of adult females (Pearson correlation = 0.76; RSME = 6.9; Fig. 3B) and better predicts their temporal pattern (Figs. 3C and 3D). Results of the simulation study indicated that 10 traps were sufficient to give 80% power in detecting the mean

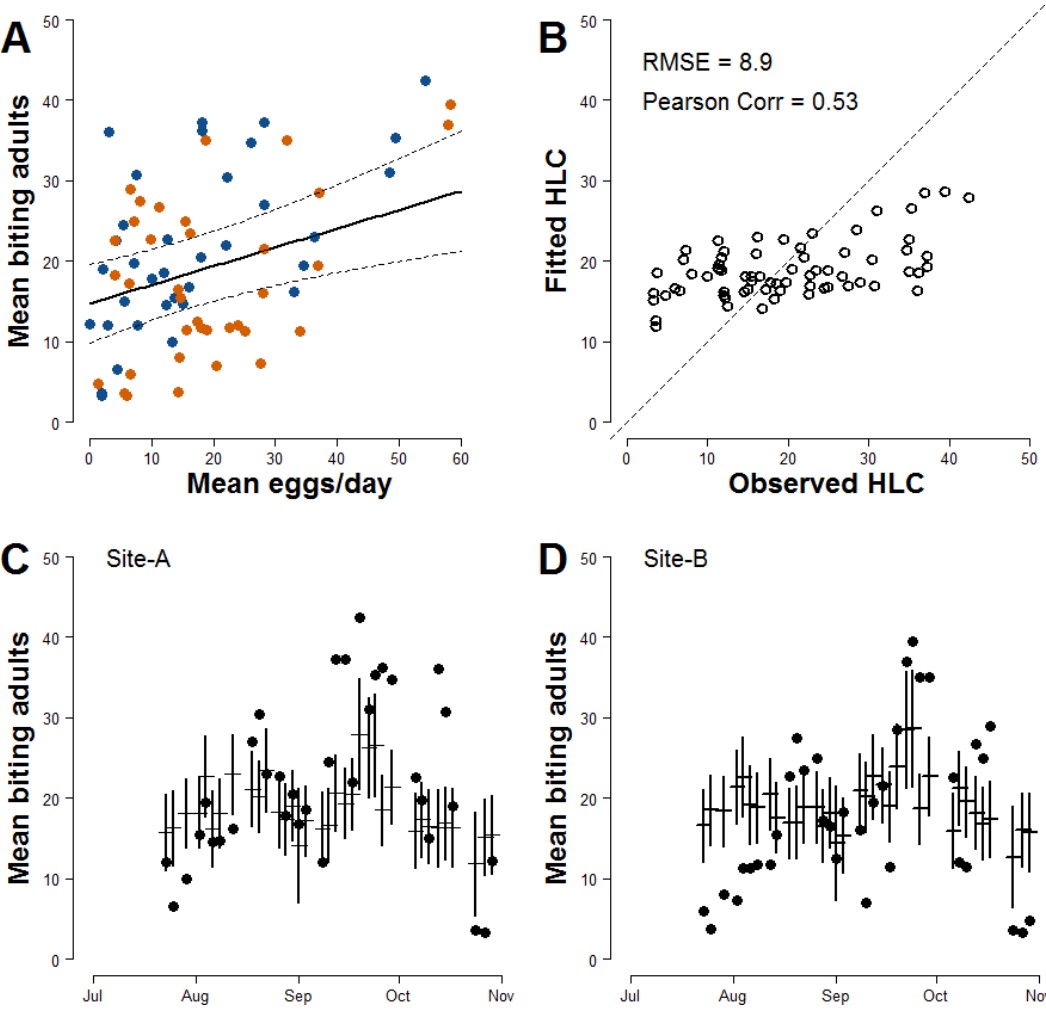

**Figure 2  Basic relationship between ovitrap collections and HLC (Model-I).** (A) *x*-axis, mean number of eggs/site/day; *y*-axis, mean number of *Ae. albopictus* biting females. Solid line, fitted values; dashed lines, 95% confidence intervals; blue dots, Site-A observed data; orange dots, Site-B observed data. (B) Observed *vs* Fitted HLC values. (C) Site-A observed and fitted values of the mean number of biting females collected during HLC along the season. *x*-axis, date of collection; *y*-axis the mean number of biting females; horizontal mark, fitted values; dark dots, observed data; vertical solid lines, 95% confidence intervals. (D) Site-B observed and fitted values of the mean number of biting females collected during HLC along the season. *x*-axis, date of collection; *y*-axis the mean number of biting females; horizontal mark, fitted values; dark dots, observed data; vertical solid lines, 95% confidence intervals.

number/eggs/day effect and that a further increase of the number of ovitraps would have a low probability to improve the results (Fig. S2).

## Estimates of risk of exotic arbovirus autochthonous transmission

Estimates of $R_0$ for CHIKV in the study area range from 1 to 2.4 when calculated both on the basis of observed and fitted biting females, with the exception of few dates at the beginning and at the end of the sampling period (Fig. S1). On the contrary $R_0 < 1$ is always obtained for DENV and ZIKAV, with the exception of few sampling dates between late August and October, when $R_0$ for ZIKAV ranges between 1 and 1.5 (Fig. S1). Figure 4

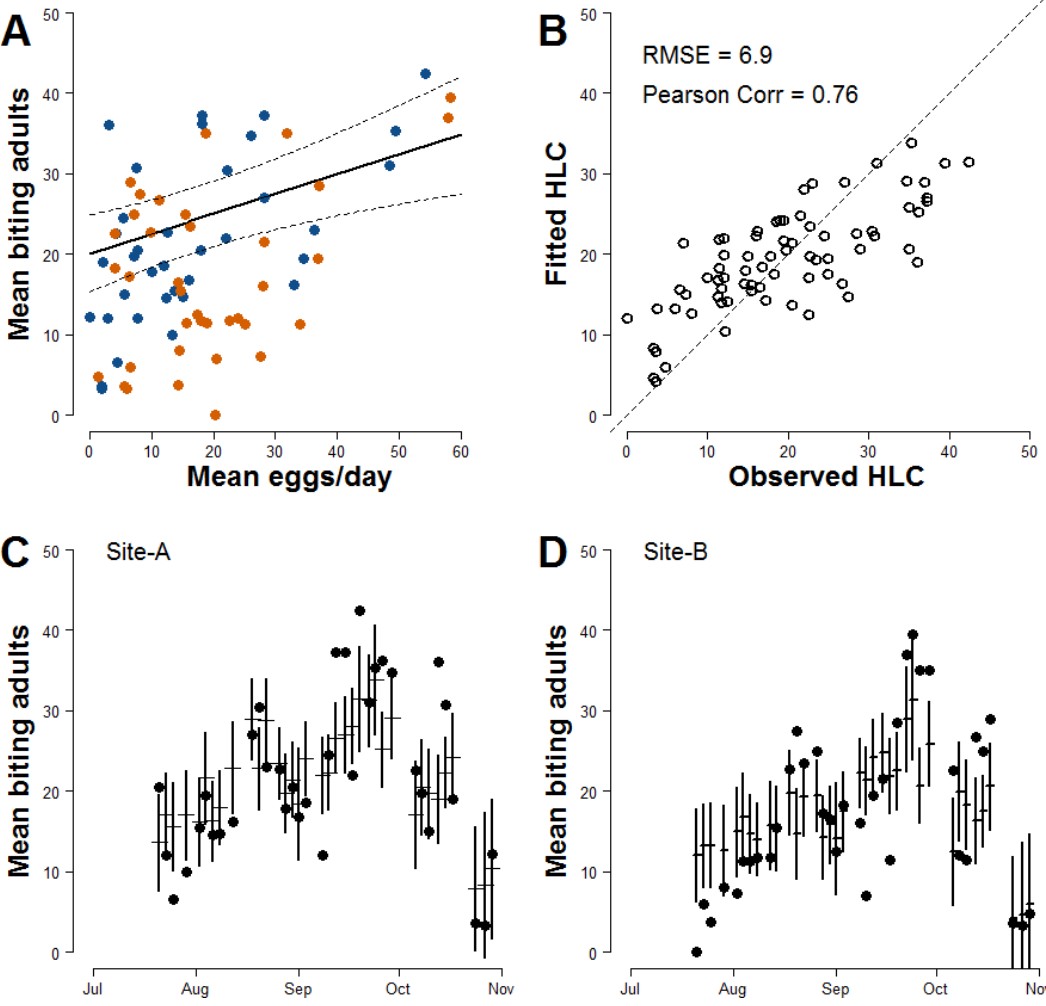

**Figure 3** **Improved relationship between ovitrap collections and HLC (Model-II).** (A) *x*-axis, mean number of eggs/site/day; *y*-axis, mean number of *Ae. albopictus* biting females. Solid line, fitted values; dashed lines, 95% confidence intervals; blue dots, Site-A observed data; orange dots, Site-B observed data. (B) Observed *vs* Fitted HLC values. (C) Site-A observed and fitted values of the mean number of biting females collected during HLC along the season. *x*-axis, date of collection; *y*-axis the mean number of biting females; horizontal mark, fitted values; dark dots, observed data; vertical solid lines, 95% confidence intervals. (D) Site-B observed and fitted values of the mean number of biting females collected during HLC along the season. *x*-axis, date of collection; *y*-axis the mean number of biting females; horizontal mark, fitted values; dark dots, observed data; vertical solid lines, 95% confidence intervals.

shows the relationship between the mean number of eggs/site/day and the values of $R_0$ for CHIKV computed using average HLC values (solid lines) with their confidence intervals (grey area) predicted by Model II during *Ae. albopictus* reproductive season (from June to September). Despite the large confidence intervals in the estimation of $R_0$ values for CHIKV based on fitted biting females, results indicate that $R_0$ is >1 when at least 28, 20, 20, 3, 12 and 79 eggs/day are collected between June and November, respectively. Below these numbers of eggs/day, $R_0 = 1$ is included within the confidence intervals and does not allow to predict the onset of the outbreak with 95% of confidence. Similar patterns of the

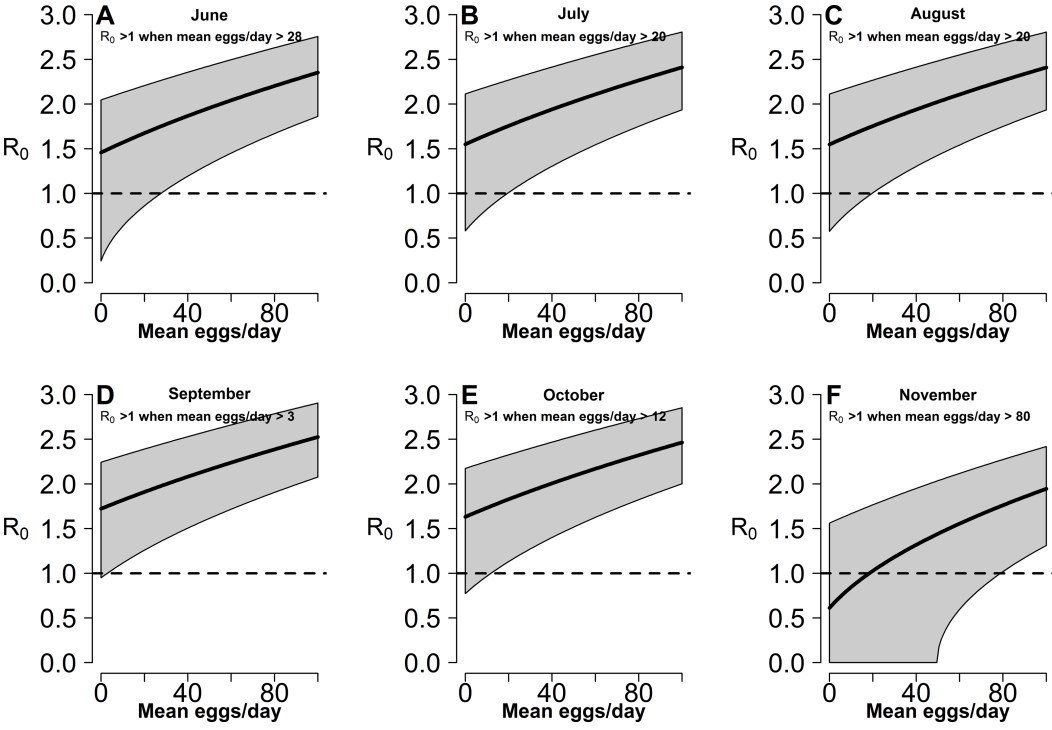

**Figure 4** **Relationship between mean number eggs/day/ovitrap and $R_0$ estimates along *Aedes albopictus* reproductive season in a highly infested area in Rome.** (A–F): $x$-axis, mean eggs/day; $y$-axis, mean $R_0$. Solid black line, mean $R_0$ value computed using average HLC values predicted by Model II for the given value of mean eggs/day. Grey area, 95% confidence intervals. Meteorological variables were considered at their monthly mean values. (A) June, (B) July, (C) August, (D) September, (E) October, (F) November.

risk of outbreak for arboviruses in the study area are obtained either based on HLC data or on estimates of biting females from Model-II (Fig. 5). Risk of CHIKV outbreak ranges from 40 to 80% from the second half of August to the end of the October, with only few exceptions Figs. 5A and 5B). Risk of ZIKAV ranges between 0 and 20% up to second half of September when it raises up to 40% and decreases afterwards (Figs. 5C and 5D). No risk of outbreak ($p = 0$) is predicted for DENV (not shown).

## DISCUSSION

Ovitrap data are considered appropriate to assess presence/absence of *Ae. albopictus* in a given site but not adult abundance, due to the several biases potentially affecting the outcome of ovitrap collections and their relationship with the adult mosquito population (*Qiu et al., 2007*; *Straetemans, 2008*; *ECDC, 2012*). However, due to feasibility and economic reasons, the number of eggs in ovitraps represents the most commonly available data provided by large-scale routine monitoring activities carried out by public administrations in infested areas, at least in Europe (e.g., *Severini et al., 2008*; *Carrieri et al., 2012a*; *Flacio et al., 2006*; *Collantes et al., 2016*). Thus, number of eggs in ovitraps is often taken as the only indicator of high nuisance or of higher risk of disease transmission and used for planning mosquito control interventions. Establishing a threshold in the number of eggs/ovitrap

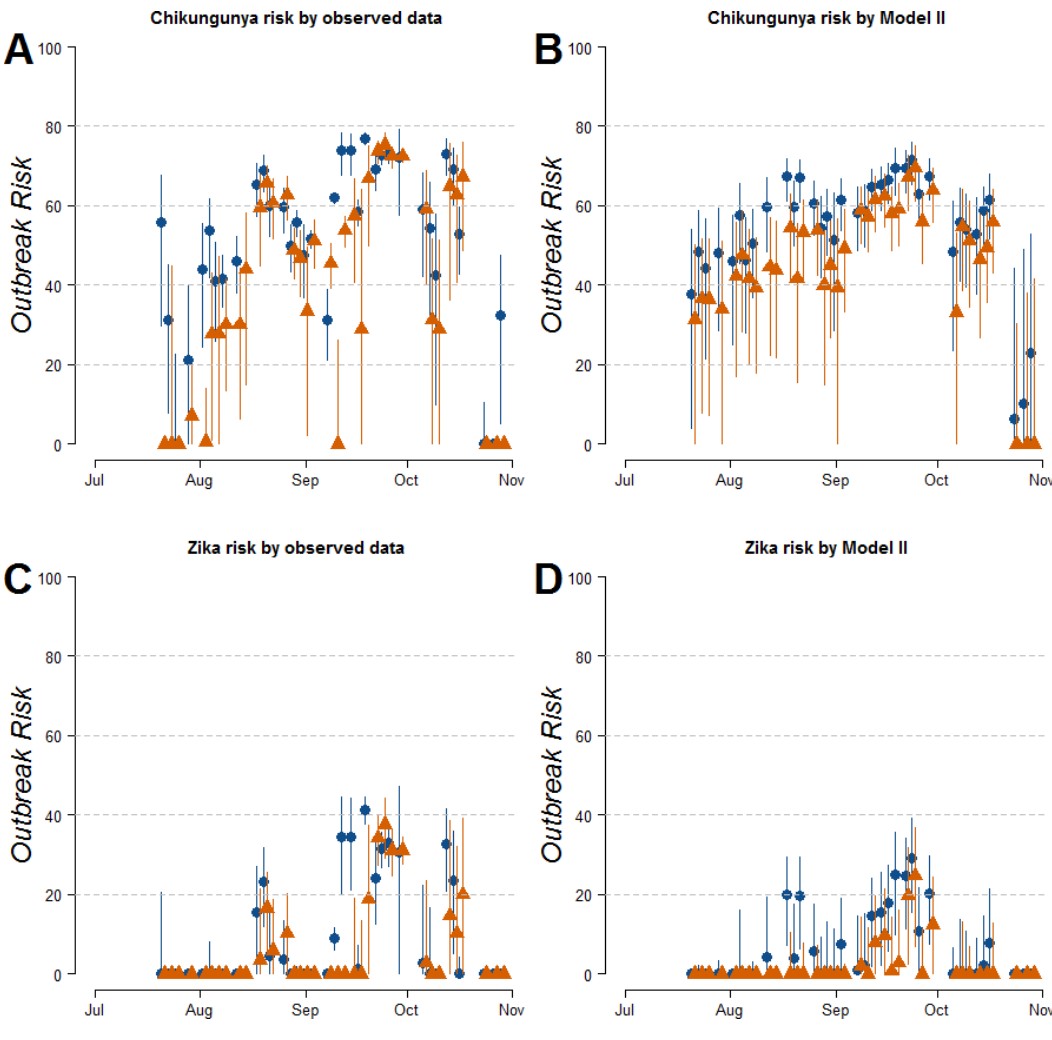

**Figure 5 Estimated risk of exotic arbovirus outbreaks from an infected case in a highly infested area in Rome.** Estimated risk of exotic arbovirus outbreaks based either on observed HLC data (A, B), or on the mean number eggs/site/day and its estimated relationship with biting *Ae. albopictus* females by Model-II (C, D). *x*-axis, months; *y*-axis, outbreak probability. Blue dots, mean values in Site-A; Orange triangles, mean values in Site-B; solid lines, confidence intervals.

over which nuisance could affect the quality of life (*Halasa et al., 2014*) and represent a risk of arbovirus transmission could serve as a very useful tool for decision-makers in charge of planning mosquito-control activities in infested areas.

This work provides the first evidence of a significant positive relationship between ovitrap data and data from HLC, i.e., the gold standard for assessing biting rate of human-biting mosquito (*Silver, 2008*) and estimating nuisance and risk of arbovirus transmission.

Results also highlight the possibility to predict mean number of adult biting females based on mean number of eggs. Counterintuitively, the mean number of eggs at Lag 0 provided a better fit than the lagged effects. Indeed, eggs have a double significance: they may reflect either eggs from which the collected adults were originated (Lag 1 and 2) or eggs laid by

collected adults (Lag 0). The reason why the latter provided the best fit may be that blood-feeding follows oviposition in a short time. This would imply that the number of biting females is correlated with those of ovipositing females in few previous days. On the other hand, larval development is more affected by climatic conditions over a long time and the relationship with production of adults eventually seeking for host is likely to change along the season, weakening the significance of Lag 1 and 2. In addition to this, it is likely that the same climatic conditions affect in the same way oviposition and host-seeking behaviours of the population at a given time strengthening the effect of Lag 0. In order to improve the prediction, several variables are considered: daily temperature, daily wind speed and the lagged effect of rainfall, reflecting the negative effect of not-optimal temperatures, of strong winds and of precipitation on adult mosquito flight and survival (*Hawley, 1988*; *Waldock et al., 2013*). However, despite this significant relationship, the accuracy of the prediction is relatively low, as indicated by wide confidence intervals on the predicted values (e.g., for a prediction of 20 females, the observed value is predicted to be between 6 and 34 in 95% of the cases). This low accuracy was expected due to the several local eco-climatic factors potentially affecting mosquito biting and oviposition activities, as well as to possible migration from neighbouring areas and the experimental scheme adopted. In particular, it should be noted that in the present work, a 15′-long HLC on unprotected volunteers in the daily peak of *Ae. albopictus* activity (*Hawley, 1988*; *Delatte et al., 2010*; *Carrieri et al., 2012b*) was taken as a proxy of the number of biting female/person/day. Moreover, the competition of other human hosts present during the HLC and of natural oviposition sites alternative to ovitraps were not taken into account.

Model prediction accuracy is also affected by sampling effort; on one hand, increasing the number of traps would decrease uncertainty of model prediction, on the other hand, at small scale as in our experimental design, an intensive sampling effort could affect mosquito population dynamic. Here we detect that our choice of using 15 traps well compensate both aspects, in fact power analysis (Fig. S2) indicates that 15 traps are sufficient to have a good statistical power (higher than 80%) but are negligible compared to the number of natural breeding sites in the study sites (botanical and enclosed gardens).

In the study area, the models predicted an increase of one biting female/person every 5 additional eggs found in ovitraps, possibly reflecting that each female had a high number of oviposition sites alternative to ovitraps where to lay its eggs, consistent with the species skip-oviposition behaviour (*Hawley, 1988*; *Davis et al., 2015*; *Davis, Kline & Kaufman, 2015*). The models estimated the presence of adult biting females also at zero mean number of eggs/day, as also observed during the experiment. This is counterintuitive, as each adult female releases tens of eggs each gonotrophic cycle, and questions the widely accepted concept that ovitraps are a very sensible tool to detect the presence of adult females.

From the epidemiological perspective, the observed number of biting female/person was in the range of those estimated in Emilia Romagna during the 2007 CHIKV-outbreak (*Poletti et al., 2011*) and of those observed in other north-east Italy sites (*Marini et al., 2015*), where similar models predicted a non-negligible risk of exotic arbovirus outbreaks (*Guzzetta et al., 2016a*; *Guzzetta et al., 2016b*). Risk models predicted that the extremely high biting rates observed in the study area were associated to an $R_0 > 1$ along most of the season for

CHIKV and in only a few weeks during the peak of mosquito abundance for ZIKAV. It is interesting to note that risk models also showed that risk of CHIKV and ZIKAV outbreak was higher not only at the peak of the summer season (i.e., August), but also in October, reflecting the bimodal population dynamics already reported for the species in Rome (*Manica et al., 2016*). Notably, these patterns are not to be extended to the whole metropolitan area of Rome, as both study sites are hot-spots of *Ae. albopictus* abundance, due to the presence of small green islands within a highly urbanized environment (*Manica et al., 2016*).

When estimates of adult biting *Ae. albopictus* females based on ovitrap data were exploited in risk models, the patterns of exotic arbovirus outbreak probability were similar to those obtained based on collected adults. The model allowed to predict the dynamics of the risk of arbovirus outbreak in the study area based on the number of eggs in ovitraps and to obtain threshold values of mean number of eggs/day above which interventions to prevent the transmission need to be implemented. For example, in the case of CHIKV, which had the highest outbreak probability, mean numbers of eggs/ovitrap/day ranging from three to 20 were associated to actual risk of transmission from June to October. This range is frequently observed in Rome (*Di Luca et al., 2001*; *Toma et al., 2003*), suggesting that the city has high risk of CHIKV outbreak in the presence of infected human hosts. However, it remains to be established whether the relationship between eggs and biting adults is maintained also in areas less suitable for high mosquito densities than the study sites.

The models here applied to estimate adult biting *Ae. albopictus* females based on ovitrap data could be further improved by introducing other variables (e.g., number of oviposition sites alternative to ovitraps) or by a more intense sampling effort with ovitraps, thus resulting in more accurate epidemiological estimates. However, the results here obtained represent a caveat regarding the significance of relying on large scale ovitrap monitoring schemes for estimating numbers of biting females and planning control interventions aiming at preventing risk of arbovirus transmission (or of high nuisance). In order to fill the gap between entomological studies, operational field surveillance and planning of mosquito control activities, efforts should be concentrated on the development and validation of new strategies to predict risk of arbovirus outbreaks and possibly provide straightforward warning thresholds.

## ACKNOWLEDGEMENTS

We are very grateful to Antonello D'Alessandro e Giovanni Marafini for their major contribution to human landing collections.

### Funding

This work has been funded by EU grant FP7-261504 EDENext, and is catalogued by the EDENext Steering Committee as EDENext475. RR was partially funded by the Autonomous Province of Trento (Italy), Research funds for Grandi Progetti, Project LExEM (Laboratory of Excellence for Epidemiology and Modelling, http://www.lexem.eu). The funders had no

role in study design, data collection and analysis, decision to publish, or preparation of the manuscript.

### Grant Disclosures
The following grant information was disclosed by the authors:
EU grant: FP7-261504 EDENext.
Autonomous Province of Trento (Italy).
Research funds for Grandi Progetti.
Project LExEM.

### Competing Interests
The authors declare there are no competing interests.

### Author Contributions
- Mattia Manica conceived and designed the experiments, analyzed the data, wrote the paper, prepared figures and/or tables.
- Roberto Rosà analyzed the data, reviewed drafts of the paper.
- Alessandra della Torre conceived and designed the experiments, contributed reagents/materials/analysis tools, wrote the paper.
- Beniamino Caputo conceived and designed the experiments, performed the experiments, analyzed the data, contributed reagents/materials/analysis tools, wrote the paper.

### Data Availability
   The raw data has been supplied as a Supplementary File.

### Supplemental Information
Supplemental information for this article can be found online at http://dx.doi.org/10.7717/peerj.2998#supplemental-information.

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
