# Peer review of "From eggs to bites: do ovitrap data provide reliable estimates of Aedes albopictus biting females?"

_PeerJ, doi:10.7717/peerj.2998_

## Round 0.1 · original submission · Major Revisions

· Academic Editor

Major Revisions

First, both reviewers expressed concerns about the clarity of the text, mainly regarding the English language; therefore I suggest a revision by a mother-tongue speaker.

In addition, several issues, mainly concerning the experimental design, should be addressed by the authors. Therefore at present the article needs substantial revisions.

Reviewer 1 ·

Basic reporting

Clarity could be improved, parts are hard to follow and relevant literature could be covered more directly.

eg
Abstract
“Wide prediction intervals….” This sentence is not particularly clear. Prediction of what?
“that under experimental conditions….” Again this is not clear, what are the experimental conditions?
The last sentence could be improved. So what should monitoring be based on then? A combination of adults and larvae?

Introduction
58-9 “arboviruses”. Something missing after “among which”.
63 “…e…”?
Mention similar studies in your introduction that have looked at this issue. For instance, Tantowijoyo et al (J Medical Entomol 2016, v 53) looked at a similar issue and there are other papers. The question and approach are not new.
86 Perhaps state instead that you are investigating whether there is a relationship rather than “finding” one.
90 I’m not sure you’d ever expect to relate these data directly to outbreaks which depend on other factors like the presence of the virus in the first place. I realize that risk is being considered but this is not always clear.
92 Point out that this is variation in time and not space that is being assessed.

Discussion
298-300. I’m not sure you’d call them the gold standard. It depends on where you are in the world as to what the best approach is considered to be. For instance, in some locations BG traps or gravid traps are favoured.
314 In relation to this issue is it possible to estimate how much improvement might be gained from increasing the number of ovitraps? I find these to be quite low given the known variance in numbers that can occur in a trap.
318-319. In aegypti the skip oviposition model is now seen as not accurately reflecting mosquito behaviour – because related individuals are often collected from the same ovitrap/breeding site. I wonder if the same situation exists or albopictus.

Experimental design

I have some comments.
108 Not sure what is meant by “3 times/weeks”.
118-9 Were any mosquitoes missed in this procedure? (not killed and not collected?)
125 Why the different numbers in the two sites?
I don’t really understand how the state of the water in containers was controlled if ovitraps were repeatedly used.
128. What does “some eggs” mean?
129-130 2-4 days is quite variable.
141. What does a dichotomous index mean?
143 “prior to” not “previous”
145. Change headings so this refers to the entire section on analysis.
170. missing comma after “available”
I’m not sure if the “basic” and “improved” division is useful here, why not just run these together in one section? Given the large number of variables it would be useful to know how many data points (events) were available for these analyses.
216 Use subscript 0 in R0, and check rest of paper.

Validity of the findings

I have some comments.
242. One of these values is not <.5.
Fig. 1 Why draw a line between points for one site and not the other for ovitrap data?
Fig. 2. I find these consistent departures from the dotted line in B a little odd given the data presented in Fig 1. which indicate some high values below the line.
Not sure you need “Year 2014” in the x axis.
Regression line confidence intervals. These refer to the estimated slope, right? Rather than the points themselves, I suspect this should be spelt out.

Additional comments

No further comments.

·

Basic reporting

Being a non-native English speaker, I understand the challenges of writing papers in English. I have to, however, point to the fact that there are numerous basic errors in grammar throughout the text. Also, examples of convoluted long sentences can be found in the opening Abstract and Introduction paragraphs. I had difficulty following the Introduction and Discussion in particular, and I suggest their thorough editing. The general structure of the text, general rationale for the study and referencing are fine.

Experimental design

There are several points of the experimental design I feel should be explained/defended:
1) Why were females killed? Are you sure that their killing upon landing did not significantly influence the subsequent data collection...In other words, by collecting the HLC data this way, did you confound the time series you measured?
I’m assuming the effect of removing the measuring points (females) can be small in an area of high mosquito density.......But I could not find any estimate of the total (census) population size....Was this your rationale? If so, this needs to be clearly stated in your M&M.

2) Continuing on the previous issue – if you are removing females that just bloodfed, they will not leave any eggs.....over time, the very relationship that you’re trying to infer would weaken....Again, you need to explain how you think this is not an issue in your experimental data collection.

3) Why did you choose a lag of 0-14 days for the number of eggs as a predictor of the number of “currently” biting females? How long is the cycle from emergence till reproductive maturity (bloodfeeding?) Are you assuming overlapping generations? Again, none of these basic rationales are stated in your text.

4) How do you explain that the 0 lag days for the egg count was the best predictor? It is indeed biologically counterintuitive...

5) Are the study areas isolated? – How many mosquitoes are moving in/out of them? Some estimates suggest the Ae. aelbopictus flight distance to be over 400m. How can immigration of already bloodfed females (that only oviposit in your study are) affect your results?

6) Why did you choose five ovitraps? How far apart were they from each other and from the HLC data collector?

7) How do you make sure you did not over-parametrize your models?

Validity of the findings

Until the stated questions around the experimental design rationale, data collection and modeling are addressed, I cannot reliably assess the validity of the findings.

Additional comments

Manica and colleagues report on the attempt to model the number of biting Aedes albopictus females from the number of eggs collected over the entire reproductive season in two areas of high mosquito density in a metropolitan area (Rome, Italy). Additionally, they use the observed and expected number of biting females to estimate the risk of autochthonous transmission for several exotic arboviruses. They conclude that the presented approach to find an easily collectable proxy for the number of biting females is not overly precise.
I raise several major issues with the experimental design and therefore recommend Major revision.

End of the comments.

Respectfully submitted,
Gordana Rasic

---

## Round 0.2 · Minor Revisions

· Academic Editor

Minor Revisions

Dear Dr Caputo,
your manuscript has been substantially improved and now addresses most of the concerns expressed by the reviewers. In my opinion it needs just a few minor revisions prior to acceptance.

·

Basic reporting

significantly improved

Experimental design

valid and more thoroughly presented

Validity of the findings

limitations and caveats are now clearly stated

Additional comments

Manuscript has now been substantially improved. Authors have thoroughly addressed the questions raised by the reviewers, particularly the rationales for chosen approaches/parameters and much clearer statements on the limitations/potential caveats. I have no problem recommending the minor revisions of this manuscript version.
Just two points:
(i) Instead of having a title as a question – you could change it to a confirmation statement (e.g. From eggs to bites: a limited utility of ovitrap data for estimating biting Aedes albopictus females).
(ii) I think it is worthwhile including the simulation (Model-II) details/inferences into the manuscript.
Regards,
Gordana

---

## Round 0.3 · accepted · Accept

· Academic Editor

Accept

I think that the manuscript is now ready to be published. I also think that it is not necessary to change the title proposed by the authors, since the new title does not add any added value to the article.